# Remote sensing reveals fire-driven enhancement of a C$_4$ rhizomatous alien grass on a small Mediterranean volcanic island

Riccardo Guarino[1]*, Daniele Cerra[2]*, Renzo Zaia[3], Alessandro Chiarucci[4], Pietro Lo Cascio[5], Duccio Rocchini[4], Piero Zannini[4], Salvatore Pasta[6]

[1]Department of Biological, Chemical and Pharmaceutical Sciences and Technologies (STEBICEF), University of Palermo, 90123 Palermo, Italy
[2]Remote Sensing Technology Institute (IMF), German Aerospace Center DLR, 82234 Oberpfaffenhofen, Germany
[3]Magmatrek, 98050 Stromboli (ME), Italy
[4]BIOME Lab, Department of Biological, Geological and Environmental Sciences, Alma Mater Studiorum, University of Bologna, 40126 Bologna, Italy
[5]NESOS, 98055 Lipari (ME), Italy
[6]Institute of Biosciences and BioResources (IBBR), National Research Council, 90129 Palermo, Italy

*These authors contributed equally to this work.
**Correspondence:** Riccardo Guarino (riccardo.guarino@unipa.it)

**Abstract.** The severity and the extent of a large fire event that occurred on the small volcanic island of Stromboli (Aeolian archipelago, Italy) on 25-26 May 2022 was evaluated through remotely sensed data to assess the short-term effect of fire on local plant communities. For this purpose, the differential Normalised Burned Index (dNBR) was used also to quantify the extent of early-stage vegetation recovery, dominated by *Saccharum biflorum* Forssk. (Poaceae), a rhizomatous C4 perennial grass of paleotropical origin. The burned area was estimated to have an extension of 337.83 ha, corresponding to 27.7% of the island surface and to 49.8% of Stromboli's vegetated area. On the one hand, this event considerably damaged the native plant communities, hosting many species of high biogeographic interest. On the other hand, S*accharum biflorum* clearly benefited from fire. In fact, this species showed a very high vegetative performance after burning, being able to exert unchallenged dominance in the early stages of the post-fire succession. Our results confirm the complex and probably synergic impact of different human disturbances (repeated fires, introduction of invasive alien plants) on the natural ecosystems of small volcanic islands.

**Keywords.** Biological succession, Disturbance, Satellite imagery, Sprouters, Vegetation dynamics.

## Introduction

Wildfires are a main disturbance factor affecting the Mediterranean terrestrial ecosystems, whose vegetation patterns are largely influenced by interactions with fire. Fire frequency and severity delineates landscape attributes (Pausas, 2006; Jouffroy-Bapicot et al., 2021), affects the structure and composition of the vegetation (Trabaud, 1994) and regulates speed and direction of ecological succession dynamics (Canelles et al., 2019). Also, fire causes sudden variations in the carbon and energy balance of ecosystems (Novara et al., 2013; Harris et al., 2016; Pausas & Millán, 2019) and in the soil microbial activity and functional diversity of the microbiome (Velasco et al., 2009; Goberna et al., 2012).

At the onset of human civilisations, Mediterranean landscapes have been deeply modified by anthropogenic fires that were used to expand the open-canopy space available for human activities and facilitate a wide array of foraging activities (Pausas and Keeley, 2009). Throughout human history, demographic fluctuations, innovations and cultural exchanges have always been accompanied by changes in land use and thus in fire regimes, amount and patchiness of fuel (Guyette et al., 2002; Driscoll et al., 2021).

After the mid-20th century, land abandonment associated with an increase of woody cover and the build-up of fuels (Mantero et al., 2020) chiefly contributed to the increased fire hazard in the Mediterranean Region (Le Houérou, 1993; Salis et al., 2022). Despite the occurrence of some natural factors favouring fires, most of them are ignited by men through carelessness or voluntary action. Being the vegetation burning strongly related to plant water content (Bond and Wilgen, 1996), fires happen mostly during the warmest and driest months, i.e. during the Mediterranean summer (Bergmeier et al., 2021). Climate change scenarios indicate rising temperatures and decreasing amounts of precipitation, resulting in longer summer aridity, soil water shortages and increasing fire risk (Moriondo et al., 2006; Lozano et al., 2017; IPCC, 2021), despite lower productivity may limit fuel availability (Baudena et al. 2020).

Nevertheless, typical Mediterranean shrublands are highly resilient to relatively frequent, high-intensity fires, but changes in the fire regime may make these communities susceptible to compositional changes, potentially followed by alien plant invasions (Keely and Brennan, 2012; Vallejo et al., 2012). The positive feedback between invasive species and fire can be a major cause of unidirectional change in invaded ecosystems (Brooks et al., 2004), and invasive species able to sustain an increased fire frequency and intensity may generate favourable conditions for their self-perpetuation (Pauchard et al., 2008).

Small islands are particularly vulnerable to biological invasions (Bellard et al., 2016), due to the combined effect of the reduced species pool and the competitive traits of invasive species. This process has been reported for Mediterranean islands (Celesti-Grapow et al., 2016; Fois et al., 2020), particularly in the case of volcanic islands with ongoing or recent volcanic activity (Karadimou et al., 2015; Pasta et al., 2017; Chiarucci et al., 2021).

The island of Stromboli is the summit of the youngest and most active volcanic complex in the Aeolian Archipelago (NE-Sicily); its subaerial activity began around 85 ka BP (Francalanci et al., 2013) and the emerged part consists of a single cone rising up to 926 m above sea level. Stromboli has the lowest number of species, as expected by the within archipelago species-area relationship among the seven largest islands of the Aeolian Archipelago, both for native and alien species (Chiarucci et al., 2021). By far the most common invasive alien species in Stromboli is *Saccharum biflorum* Forssk. [= *S. spontaneum* L. subsp. *aegyptiacum* (Willd.) Hack.; henceforth: *Saccharum*], a is a vigorously growing rhizomatous grass of Palaeotropical origin (Amalra and Balasundaram, 2006) with culms 1.5-2.5 m and flowering stems up to 3 m high. Its rhizomes can be up to 6 m long, with nodes every 10-15 cm, from which the culms and fascicled roots branch off (Supplement 1, Fig. S1). This species has a $C_4$ metabolism and thrives in sandy-silty, often alluvial soils (Pignatti et al., 2017-2019).

*Saccharum* was introduced in the 19th century as a windbreak. Gussone (1832) recorded its occurrence (despite wrongly identifying it as *Saccharum ravennae* L.) on the islands of Stromboli, Panarea, Lipari and Vulcano, as "cultivated hedges in vineyards". *Saccharum* has then spread on former cultivations, abandoned terraced fields and wherever there was accumulation of volcanic ash, as noticed by Ferro and Furnari (1968): "a large part of the north-eastern slope of the island, the very slope that Lojacono (1878) travelled through 'vineyards that produce beautiful wines', is covered by dense, almost monophytic *Saccharum* vegetation, from sea level up to the upper limit of the ancient crops (...). This slope could have been colonised in a different way by native floristic elements, but it is difficult to make predictions on the final outcome of the competition, given the compactness of the *Saccharum* rhizomatous apparatus".

However, photos published by Ferro and Furnari (1968) give the impression that 50 years ago *Saccharum* was more widespread than nowadays. Besides cultivation abandonment, the establishment of this plant is favoured by fire, as observed by Richter (1984). Local elder people recall a major spread of *Saccharum* soon after the fire caused by paroxysmal activity in 1930 and the subsequent abandonment of a large portion of the cultivated terraces along the eastern slopes of the island (Richter and Lingenhöhl, 2002). In following years, the spread of this species has been somewhat reduced by the development of native shrubland, which until recently was the most widespread vegetation type on the island. Another large fire event, ignited at the Punta Labronzo landfill site in 1978, promoted the recovery of *Saccharum* all over the eastern slopes above Punta Labronzo. On 25-26 May 2022, a large fire event burned much of the northern and eastern slopes of Stromboli, upstream of the villages San Vincenzo and San Bartolo. This study uses remotely sensed data to analyse the post-fire damage on local vegetation

through the application of a spectrally sensitive index, i.e. the differential Normalised Burned Index (dNBR), which has been used also to quantify the extent of the subsequent early-stage vegetation recovery, dominated by *Saccharum*, in order to highlight the ecological behaviour of this invasive alien species and its fire-driven ability to colonise new spaces.

**Material & Methods**

*Study area.* The island of Stromboli, 12.6 km$^2$, represents the northeastern end of the Aeolian Archipelago, in southeastern Tyrrhenian Sea, Mediterranean biogeographical region (Cervellini et al., 2020). The island has quite a regular slope averaging 28° and two large horseshoe-shaped flank collapses named "Sciara del Fuoco", on the northwestern-, and "Rina Grande", on the southeastern flank.

Our study area covers an area of ca 3.4 km$^2$, between 50 m a.s.l. and 530 m a.s.l., on the northern and eastern sides of the volcano and can be roughly divided in two sectors. The northern sector is bounded by the "Fili del Fuoco" ridge, overlooking "Sciara del Fuoco", to the west and by the Vallonazzo valley to the east; the eastern sector is bounded by the Vallonazzo valley to the north-west and by the "Rina Grande" depression to the south-east (Fig. 1). Both sectors are characterised by medium to gentle slopes, with 80% of the area sloping less than 30° (Fornaciai et al., 2010).

The climate of Stromboli is typically Mediterranean. At 4 m a.s.l. the average yearly temperature is 18.2 °C, with a mean temperature of 12.3 °C in the coldest (January) and 26 °C in the warmest month (August). The annual rainfall averages 570 mm, while the relative humidity is 75.0% in winter and 60.8% in summer. Based on the WorldClim interpolated maps (Hijmans et al., 2005) and on the Rivas-Martínez bioclimatic classification (2004), the study area is characterised by an upper thermo-mediterranean thermotype and a dry to sub-humid ombrotype (Bazan et al., 2015).

The study area was dominated by a typical Mediterranean rockrose garrigue (*Cistus creticus* subsp. *eriocephalus*, *C. monspeliensis*, *C. salvifolius*) with scattered patches of maquis with *Genista tyrrhena*, *Spartium junceum*, *Olea europaea*, *Erica arborea* and *Pistacia lentiscus* (Richter, 1984; Cavallaro et al., 2009). The former cultivated land and the volcanic ash deposits were extensively colonised by *Saccharum*, while small *Quercus ilex* stands were occasionally found along the impluvium lines. Equally rare and scattered were the patches dominated by *Euphorbia dendroides*, limited to the rocky outcrops, especially along the south-facing rim of Vallonazzo valley (Ferro and Furnari, 1968; Richter and Lingenhöhl, 2002). The highest and southernmost end of the study area included part of the local population of *Cytisus aeolicus*, a narrow ranging endemic broom growing only on the islands of Vulcano, Alicudi and Stromboli (Zaia et al., 2020).

On 25-26 May 2022, due to recklessness during the filming of a television drama, a fire broke out in the upper outskirts of the village of San Vincenzo and, fuelled by a strong sirocco wind, burned the whole of our study area. While *Saccharum* stands were entirely burned, a very few small patches of garrigue and *Quercus ilex* stands escaped from the fire.

*Satellite imagery processing.* To infer the extent of fire damage to the vegetation and the post-fire surface of the resprouted *Saccharum* patches, we used optical satellite images acquired by the spaceborne Sentinel-2 sensor, a multispectral mission launched in the frame of the European Space Agency (ESA) Copernicus program (Drusch, 2012).

Sentinel-2 measures globally the backscattered solar radiation from ground targets with a temporal resolution of around 5 days, across 13 spectral bands with different ground sampling distance (GSD) varying from 10 to 60 metres. In this work, we employed the four bands at 10 m GSD, namely in the visible range (blue, green, red) and near infrared (NIR). Additionally, we relied on Band 12 in the short-wave infrared (SWIR) at 20 m GSD in order to detect burned areas. Additionally, spectral bands 5, 6, 7, 8a, and 11, all at 20 m GSD, were used for the supervised classification of different vegetation types. All other bands at 60 m GSD were not used in this analysis. The products used were at processing level 2A, which provides radiometrically corrected, georeferenced, orthorectified, atmospherically corrected, and converted to bottom of atmosphere reflectance data. The choice of using reflectance rather than radiance products is motivated by the following reasons: (1)

overall brightness differences in different images due to different acquisition conditions are reduced in the level 2A products, (2) quantities estimated from single images through spectral indices result more meaningful when applied to data in reflectance. The data selection and processing were carried out on Google Earth Engine (GEE) (Amani et al., 2020), which is at the same time a multi-petabyte repository of geo-referenced and harmonised Earth Observation raster, vector, and tabular datasets, which includes the whole Sentinel-2 archive.

To quantify the damage caused by the above mentioned fire event on the vegetation, different Sentinel-2 scenes acquired in a relatively short time span were aggregated. An image composite of the island before the event was derived by considering 8 acquisition dates with cloud cover below 5% acquired before the fire event, from April 15 to May 22, 2022, and considering the median reflectance for each image element. This allows removing abnormal values due to specific atmospheric conditions inducing error in the reflectance estimation process, undetected clouds, and cloud shadows in the scene. The post-fire reflectance was estimated by applying the same processing to 6 acquisition dates after the event, from May 26 to June 15, 2022. The two image composites are reported in Fig. 2. Therein, pre- and post-event true colour images obtained from Sentinel-2 bands in the visible range (namely bands 4, 3, and 2) can be visually assessed, with damage caused by the fire in the northeastern part of the island already evident in this band combination.

In order to estimate vegetation loss and total burned area, we derived the Normalised Burn Ratio ($NBR$), defined for a multispectral image $x$ as:

$$NBR(x) = \frac{NIR - SWIR}{NIR + SWIR},$$

where $NIR$ and $SWIR$ indicate reflectance in the Near and Short-wave Infrared, represented for Sentinel-2 by the bands 8 and 12, respectively. The $NBR$ is a commonly used index to detect burned area and burn severity (Key and Benson, 1996), and is particularly sensitive to the changes in the amount of live green vegetation, moisture content, and some soil conditions which may occur after fire (Lentile et al., 2006).

Change detection relying on spectral indices from multitemporal pre- and post-fire images can be used to estimate vegetation loss or recovery. Relying on the availability of multitemporal images, we used the differenced $NBR$ ($dNBR$) since it performs well in capturing the spatial severity within fire perimeters (Picotte and Robertson, 2010; Soverel et al., 2010).

The $dNBR$ related to pre- and post-event images, respectively $x_{t0}$ acquired at time $t0$ and $x_{t1}$ acquired at time $t1$, is the delta of the two measurements:

$$dNBR(x_{t0}, x_{t1}) = NBR(x_{t0}) - NBR(x_{t1})$$

This quantity has been used to estimate both fire severity and vegetation recovery after the fire event: a negative $dNBR$ is correlated to recovery after fires, while a positive one indicates damages, with severity proportional to the $dNBR$ value.

We first estimated the area affected by fire immediately after the event by computing the $dNBR$ for the whole island. The affected area was derived by applying the damage classes defined in (Key and Benson, 1996). In particular, the value of $dNBR$ in the middle of the range related to low-severity damage (0.1-0.27) and approximated to the second decimal digit, in the specific 0.19, was selected and assessed using expert knowledge in order to exclude false positives from the estimation and perform further analysis only on relevant image elements, considering damaged all image elements with $dNBR$ above this threshold (Fig. 2). This was necessary as using the value of 0.1 was raising false alarms, most notably within urban areas.

To check whether the severity of the damage was related to geomorphological features, rather than to different vegetation units, the correlation between results of the $dNBR$ and a digital elevation model (DEM), was evaluated. The Normalized Difference Vegetation Index ($NDVI$; Gandhi et al., 2015) was also applied to estimate the loss in live green vegetation, and its correlation with $dNBR$ values was checked (Supplement 2).

Finally, to evaluate the quality of our results, we computed a new $dNBR$ between the pre-event image and a mosaic of Sentinel-2 acquisitions from the time range 15-17 August 2022. The burned area detected in such way was compared with very high-resolution images acquired by a drone DJI Phantom 3 professional on 17 August 2022, i.e. around 3 months after the fire event and 5 days after the first intense rainstorm. Drone images were merged and geo-referenced through the software Agisoft

Photoscan Professional (version 1.2.6). These images have 10 cm GSD and have been mosaicked over the north-eastern part of the island, covering the inhabited area of San Bartolo and San Vincenzo. The drone images did not cover the higher elevations of our study area, closer to the volcano's vents, nor the northernmost part, near Punta Labronzo (Fig. 4).

*Vegetation recovery assessment.* The mentioned image composite of Stromboli derived from 8 acquisitions from April-May 2022 was also used to map the structural types of the vegetation affected by the fire, through supervised classification based on spectral information. Three vegetation classes have been defined: maquis, garrigue, and saccharum. The class "maquis" groups tall woody vegetation patches, namely: (1) shrublands with *Genista tyrrhena*, *Spartium junceum*, *Erica arborea* and *Pistacia lentiscus*, (2) abandoned olive groves invaded by *Cytisus infestus* and *C. laniger*, (3) *Quercus ilex* groves, (4) *Euphorbia dendroides* shrublands, and (5) *Cytisus aeolicus* shrublands. The class "garrigue" refers to vegetation patches with dwarf shrubs, subshrubs and bunchgrasses, including (1) dwarf shrublands dominated by *Cistus sp. pl*., (2) herbaceous-chamaephytic vegetation dominated by *Cymbopogon hirtus*, *Oloptum miliaceum*, *Centranthus ruber*, *Jacobaea maritima* subsp. *bicolor* and *Scrophularia canina*, (3) small impluvia colonized *Rubus* sp. and *Pteridium aquilinum*. Finally, the vegetation patches dominated by *Saccharum* were attributed to the "*Saccharum*" class, easily recognized by its typical yellowish-green colour and remarkable structural homogeneity, due to one single species covering well over 80% of the soil. These patches have two different textures: smoother where *Saccharum* has invaded abandoned vineyards, more granular where Saccharum has invaded former fig tree plantations, as it happened in the upper part of our study area.

For each of the three classes described above, 10 patches of 50 pixels each were selected by experts to constitute the training dataset and 150 random points equally split among the three classes constituted the validation dataset. The area where damage occurred was fed to a Support Vector Machine (SVM) classifier (M.A. Hearst et al., 1998), as implemented in the *libsvm* routine in GEE, using a linear kernel and setting the cost $C$ to 1. The input parameters were all Sentinel-2 spectral bands having a Ground Sampling Distance of 10 or 20 meters, namely bands 2 to 8, 8a, 11, and 12. The results of the classification algorithm (Fig. 3) were evaluated through visual analysis by the experts and numerically validated using the validation dataset, yielding an overall accuracy higher than 90%.

To check variations in the distribution of burn severity levels and to evaluate the short-term response after fire among different vegetation types, the pixel values of *dNBR* pre-post were randomly sampled in 50 random points for each of the three vegetation classes described above. Levene's test was used to assess the homogeneity of variance, followed by nonparametric Kruskal-Wallis test, using Chi-Square distribution (right-tailed) and Dunn's post hoc comparison to reject the null hypothesis. To evaluate the short-term vegetation response after fire, the composite images of Sentinel-2 acquisitions from the following time ranges were analyzed: 15-17 August 2022; 14-26 September 2022; 22-28 October 2022; 10 May-15 June 2023.

On-site surveys were carried out on 15-19 September 2022, 7-9 March and 9-12 September 2023, in order to validate the remotely sensed data and to sample vegetation plots in the burned area. The vegetation was sampled in 38 permanent plots, 10 m$^2$ each, randomly selected along a belt between 180 and 220 m elevation (Fig. 1). To optimize the sampling effort, the location of the sampling sites deviated little from the paths that run along the volcano's flank above the villages of St. Vincent and St. Bartolo. The only rules adopted were that the plots should have been at least 50 m apart, to avoid spatial autocorrelation, and that each of the above-mentioned three vegetation classes should have been represented by at least 10 plots. Vegetation data were collected using a modified Braun-Blanquet (1964) approach, by visually estimating the cover–abundance in percentage values and by measuring the mean and maximum height (in cm) of each species.

In order to collect useful information to better understand the interaction between *Saccharum* and fire, a comparative evaluation of stem density/m$^2$ in burned vs. unburned patches, was carried out in the field on 18 September 2002. Sampling plots 1 × 1 m were located every 100 m along two almost contiguous transects, 900 m long, ten inside the burned area, above the village of San Vincenzo and 10 outside the burned area, in the bottom part of Rina Grande (Fig. 1). In each plot, the number of stems of *Saccharum* was counted and the average and max. height were recorded. In the unburned patches, the relative percentage

of dry stems compared to green stems was also assessed, to showcase the ease of fire ignition due to the abundant presence of
dry biomass, consisting mainly of the flowering stems of *Saccharum* which, once faded, dry out completely but remain
standing, as they are supported by the green stems which have not yet flowered.

**Results**

The application of the *dNBR* yielded a severity map showing the difference between pre- and post-fire acquisitions. The burned
area was quantified in 337.83 ha, corresponding to 27.7% of the island surface (Fig. 2). Concerning the burn severity (Keeley
2009), 75.15 ha showed low, 218.37 ha intermediate and 44.31 ha high severity level. The Kruskal-Wallis H test indicated a
significant difference in the distribution of severity levels among vegetation classes, $\chi 2(2) = 8.56$, p = .013, having the burned
garrigue and maquis suffered higher severity damage than *Saccharum* (Fig. 3).
We found no correlation between the *dNBR* and neither the elevation nor the slope (therefore not reported here). *NDVI* values
were strongly correlated with *dNBR* values (Pearson correlation of 0.97, see Supplement 2). However, *NDVI* showed some
noise in the estimation of vegetation loss, and false positives scattered across the inhabited area. Therefore, these results are
not reported further in this paper, despite of *NDVI* having a true resolution of 10 m in Sentinel-2 products, while *NBR* employs
the SWIR band, which is originally at 20 m GSD and therefore interpolated.
Considering the limitations imposed on spatial resolution by the satellite-derived damage evaluation, the burned area detected
by *dNBR* from the mosaic of Sentinel-2 acquisitions in the time range 15-17 August 2022 matched well the burned area
observable in the drone image acquired on August 17th, with man-made structures and even single trees that were spared by
the fire correctly regarded as undamaged in the *dNBR* estimation (Fig. 4). At the same time, partially burned vegetated areas
were correctly included in *dNBR* results, because even if they did not burn completely a steep decrease in the red edge portion
of the spectrum around 700 nm revealed strong vegetative stress.
The *NDVI* calculated with a threshold of 0.08, therefore quantifying all pixels having at least 8% covered by photosynthetically
active vegetation, quantified the area of the island covered by vegetation before the fire as 678.73 ha. Considering the described
correlation between *dNBR* and NDVI, and the area affected by the fire as computed by *dNBR*, it can be concluded that roughly
half (49.8%) of the vegetated area of Stromboli has been burned during the fire event.
Figure 5 shows the vegetation recovery in the area affected by the fire. According to the thresholds suggested by Key and
Benson (1996) to categorise recovery levels from dNBR values, in the specific enhanced low and high regrowth for *dNBR*
values ranging from -100 to -250 and smaller than -250, respectively, one year after fire 53.25% of the burned area showed
high enhanced recovery, 30.84% low recovery, 15.9% no recovery. Among the three vegetation classes considered, 56.08%
of the pixels with high recovery levels were *Saccharum*, 38.2% garrigue and 5.7% maquis. Conversely, 10.46% of the areas
with no recovery were maquis, 65.48% garrigue and 23.856% *Saccharum*. Considering the distribution of recovery levels
across the first growing season after fire, *Saccharum* is clearly characterized by faster recovery with respect to the maquis and
the garrigue, particularly at the beginning of the first growing season after fire (September-October 2022).
Referring to the vegetation recovery estimated in October 2022, the Kruskal-Wallis H test indicated that there is a significant
difference among the vegetation classes, $\chi 2(2) = 8.41$, p = .015, with a mean rank score of 64.06 for *Saccharum*, 89 for garrigue,
and 73.44 for maquis. The Post-Hoc Dunn's test using a Bonferroni corrected alpha of 0.017 indicated significant differences
of *Saccharum* recovery towards both maquis and garrigue (Table 1).

**Table 1. Dunn's post hoc comparison for *dNBR*-estimated recovery of the considered vegetation classes in the burned**
**area on October 2022.**

| Pair | Mean Rank difference | Z | SE | p-value | p-value/2 |
|---|---|---|---|---|---|
| *Saccharum*-maquis | -24.94 | 2.8703 | 8.6891 | 0.004101 | 0.002051 |
| *Saccharum*-garrigue | 15.56 | 1.7908 | 8.6891 | 0.07333 | 0.03667 |
| garrigue-maquis | -9.38 | 1.0795 | 8.6891 | 0.2804 | 0.1402 |

The results of the spectral evaluation of the vegetation recovery are confirmed by the on-site surveys. Table 2 shows the median values of percentage cover and height of resprouts and seedlings in the plots sampled on September 2022, March and September 2023. The distribution of the plots across the vegetation classes was the following: 10 *Saccharum*, 16 Garrigue, 12 Maquis. The Kruskal-Wallis H test indicated highly significant differences ($p < 0.001$) between the cover values and height of resprouts and cover of seedlings in the *Saccharum* plots compared to those ascribed to the other two vegetation classes. No significant difference was found in seedlings height or even in species composition across the vegetation classes (data not shown), which in all cases was largely dominated by annual plants such as *Brassica fruticulosa*, *Ornithopus compressus*, *Lupinus angustifolius*, *Trifolium stellatum* and by seedlings of *Cistus* sp.pl. (mainly *Cistus creticus*).

**Table 2. Median values of cover (%) and height (cm) of resprouts and seedlings in the validation plots. Values in brackets indicate positive absolute deviations from the median values.**

| Date | Vegetation | Resprouts cover | Resprouts height | Seedlings cover | Seedlings height |
|---|---|---|---|---|---|
| 15-19 Sept. 2022 | *Saccharum* | 85 (5) | 150 (20) | 5 (0) | 9 (13) |
| | Garrigue | 10 (15) | 8 (17) | 25 (25) | 13 (21) |
| | Maquis | 15 (15) | 15 (12) | 30 (30) | 14 (16) |
| 7-9 March 2023 | *Saccharum* | 90 (0) | 160 (20) | 10 (5) | 43 (14) |
| | Garrigue | 20 (10) | 23 (24) | 40 (20) | 33 (22) |
| | Maquis | 20 (15) | 27 (38) | 50 (25) | 38 (25) |
| 9-12 Sept. 2023 | *Saccharum* | 90 (0) | 160 (20) | 10 (10) | 53 (19) |
| | Garrigue | 25 (15) | 20 (32) | 55 (15) | 47 (32) |
| | Maquis | 25 (30) | 36 (47) | 50 (20) | 55 (30) |

The estimated vegetation composition in the study area shows that already in August resprouting Saccharum had invaded approximately 13% of areas previously occupied by other vegetation classes, especially along gullies. This latter percentage remained almost unchanged in the following months (Fig. 6). The fast recovery of the *Saccharum* patches, with their soft green colour standing out against the surrounding black, became evident as early as a few weeks after the fire (Supplement 1, Fig.

S3-5). Until first rains, which occurred on the night of 12 August 2022, *Saccharum* was the only green spot in the fire-affected areas and the high-resolution drone images captured on 17 August 2022 clearly show all *Saccharum* patches in their recovery phase (Fig. 4). In the Sentinel2 images of September-October 2022, previous damage from the fire event appears mitigated. More in detail, a total of 110 ha of the previously burned area (roughly one third) exhibits a *dNBR* value below -0.1, which represents a strong indicator of vegetation recovery. This was mostly due to *Saccharum*, demonstrating that this species can exert unchallenged dominance in the early stages of the post-fire dynamics (succession), reaching vegetative stem densities only slightly lower than those of the unburned stands in a short time (Fig. 7).

**Discussion**

Our study confirms that fire severity can be mapped with high accuracy using indices derived from Sentinel 2 imagery with supervised vegetation classification based on spectral information (Gibson et al. 2020). Fire is a major driving force for Mediterranean insular ecosystem dynamics since the emergence of the Mediterranean climate (Médail, 2021), particularly in volcanic island ecosystems (Irl et al., 2014). This paper provides the first report of how a single fire event significantly affected Stromboli Island, burning 50% of the vegetated island surface. This clearly influenced the island biota, particularly the native vegetation, which is rich in species of relevant biogeographic interest, such as *Centaurea aeolica*, *Genista tyrrhena*, *Dianthus rupicola* subsp. *aeolicus*, *Jacobaea maritima* subsp. *bicolor* (Pasta et al., submitted). In addition, the highest and southernmost end of the study area included part of the *Cytisus aeolicus* population, one of the rarest and most emblematic endemic plant species of the Aeolian Archipelago (Zaia et al., 2020).

Although we applied a permissive threshold (8%) in the NDVI for our quantitative analysis, our conclusion that the fire occurred on 25-26 May 2022 burned roughly half of Stromboli's vegetated area appears reasonably accurate, when considering all the available data we used for validation. Our study confirms that burn severity levels, estimated by dNBR, is higher in woody vegetation (Koutsias & Karteris, 2002), presumably due to the larger above-ground biomass and dead organic matter stock in the case of maquis (Rossetti et al. 2022) and to the high flammability of Mediterranean dwarf shrubs in the case of garrigue (Dimitrakopoulos, 2001). Despite the garrigue being mostly formed by pyrophytes, obligate seeders, and among the first shrubs to emerge after fire (Palá-Paúl, 2005; Athanasiou et al., 2023), our study demonstrated that *Saccharum* exhibits even greater resilience compared to garrigue in the earliest stages after fire, with a clear risk of altering the recovery patterns of native vegetation, that especially on volcanic islands are characterized by high abundance of nitrogen fixers and annual species (Weiser et al., 2021).

The positive interaction between *Saccharum* and fire was already noticed in Stromboli by Richter (1984) and Richter and Lingenhöhl (2002). Fire spreads very easily across *Saccharum* vegetation, due to the abundant presence of standing dry biomass (Supplement 1, Fig. S2, S4, S6). This result agrees with many recent studies focused on the role of fire as promoter of $C_4$ grasses (Scheiter et al., 2012; Hoetzel et al., 2013; Ripley et al., 2015). Although the native rockrose garrigue vegetation is also adapted to - and favoured by - periodical fires (Pausas, 1999), its survival derives from the ability of *Cistus* to develop a long-lasting soil seed bank (Soy and Sonie, 1992; Scuderi et al., 2010). Too frequent fire events and runoff caused by heavy rainfall on sandy and incoherent soils may cause a critical depletion of soil seed bank and favour sprouters against obligate seeders. On this purpose, we must point out that the autochthonous sprouters (such as *Erica arborea*, *Pistacia lentiscus*, *Olea europaea*) have slower growth rate than *Saccharum* and need longer time to become established.

After the fire, our study area was exposed to full solar radiation; dark sandy surfaces were subject to extreme microclimatic (surface temperatures up to 80 °C; see Richter, 1984) and extremely dry conditions. These were not favourable for the germination of the soil seed bank, whilst sprouters faced almost no competition until first rains, which occurred on 12 August 2022. The first and most important beneficiary of these contrasting conditions was *Saccharum*, which over time was able to colonise large surfaces of tephra in the northern and eastern parts of the island, likely due to a positive interaction between land abandonment, repeated fires and volcanic ash deposition. *Saccharum* is extremely competitive thanks to a variety of

functional strategies (e.g. C4 photosynthetic pathway, large resource allocation belowground, into clonal and bud-bearing rhizomes which can boost a quick resprouting and local spread/space occupancy/resource uptake) under current and probably also under predicted conditions (likely more disturbed) which could affect and define different ecosystems on Stromboli.

According to Lojacono (1878), *Saccharum* was planted along the vineyards to shelter them from the northerly winds (Fig. 8). This condition lasted until the eruption of 11 September 1930, so far considered the most violent and destructive event in the historical records of Stromboli's activity (Rittmann, 1931). Facilitated by the winter rains and by a rapid expansion via rhizomes, *Saccharum* first benefited from the emigration of most inhabitants and subsequent abandonment of terraced fields, which in a very short time lapse were almost completely sealed off by a dense monospecific bed, which made it difficult for other species to establish themselves (Ferro and Furnari, 1968; Richter, 1984). Since then, competition for space between local native vegetation and *Saccharum* beds has been regulated mainly by the periodical occurrence of fires. Further studies are needed to understand the duration of the *Saccharum* expansion phases. Our preliminary results suggest that the expansion of *Saccharum* is surprisingly fast, but the decline may also be relatively rapid. There is no data on the longevity of *Saccharum* rhizomes and related senescence processes, nor on the effects of volcanic ash deposition on rhizome burial. However, there are reasonable indications that, if the vegetation is not too frequently affected by fire, *Saccharum* could be gradually replaced by native vegetation within a few decades, as captured in the maps published as "Fig. 4" by Richter and Lingenhöhl (2002).

On 12 August 2022, a severe thunderstorm triggered disastrous erosion processes over the entire area affected by the fire on May 25-26. Large quantities of mud, stones and volcanic ashes flooded the streets of the villages San Bartolo and San Vincenzo (Supplement 1, Fig. S7). In the burned area, the traces of runoff and surface rill erosion were still very evident during our inspections on 18-19 September 2022. However, just as evident was the ambivalent role of *Saccharum*, which, while on the one hand clearly prevails on native species, on the other hand, thanks to its dense mat of rhizomes, proves to be much more efficient than the burned native vegetation in counteracting hydrogeological instability. The latter is a very relevant aspect in a volcanic island, whose soils are largely made up of loose tephra ashes.

Over time, *Saccharum* beds have become an important secondary habitat for many animal species. In fact, they represent the main breeding site for at least 70% of breeding bird species on Stromboli (Massa et al., 2015) and host conspicuous populations of almost all terrestrial vertebrates occurring on the island (especially *Tarentola mauritanica*, *Podarcis siculus* and *Hierophis viridiflavus*). Some of the invertebrates that occurs in the *Saccharum* beds are of considerable biogeographic interest, such as *Caulostrophus zancleanus*, a regional endemic (Lo Cascio et al., 2022), and the recently described *Catomus aeolicus*, endemic of the northeastern sector of the Aeolian archipelago (Ponel et al., 2020). Although not specialised on *Saccharum*, the rhizophagous larvae of the melolonthid *Anoxia orientalis*, a species considered rare at national scale in Italy, feed on its rhizomes. Surprisingly enough, *S. biflorum* does not seem to be an attractive fodder for the mammals introduced in historical (*Oryctolagus cuniculus*) or more recent (*Capra hircus*) times, nor significant infestations of phytophagous insects have ever been observed. Thus, herbivory does not seem to be a limiting factor to the expansion of *Saccharum* on Stromboli.

**Conclusions**

Remotely sensed data provide fast, accurate and reliable information for post-fire damage analysis, being spectrally sensitive to vegetation features and structure. Multi-temporal data acquisition allows observations on early-stage vegetation dynamics which, in our case, point out the outstanding pioneer role played by *Saccharum biflorum*, showcasing its ability to colonize and dominate large areas, potentially altering the recovery patterns of native vegetation. On the other hand, *Saccharum* proves to be efficient in stabilizing the soil, especially in a volcanic island with loose tephra ashes, thus mitigating the erosion processes. Our findings underscore the complex interplay between fire, vegetation dynamics, and ecosystem recovery on Stromboli, emphasizing the need for further research to better understand the long-term dynamics of *Saccharum* expansion and its interactions with native biota.

*Author contribution.* RG and DC developed the research idea, DC processed satellite and drone imagery, RG and RZ conducted the field work, RG led the writing process, all authors discussed the results and contributed to the manuscript.

*Acknowledgements.* We would like to thank Giuseppe De Rosa, who brought DC and RG together, and Antonio Zimbone for driving the drone flight and taking the pictures used to check the quality of the information derived from dNBR analysis. Three anonymous reviewers and Gianluigi Ottaviani are gratefully acknowledges for their suggestions on an earlier version of the manuscript.

*Competing interests.* The contact authors declared that neither they nor their co-authors have any competing interests.

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

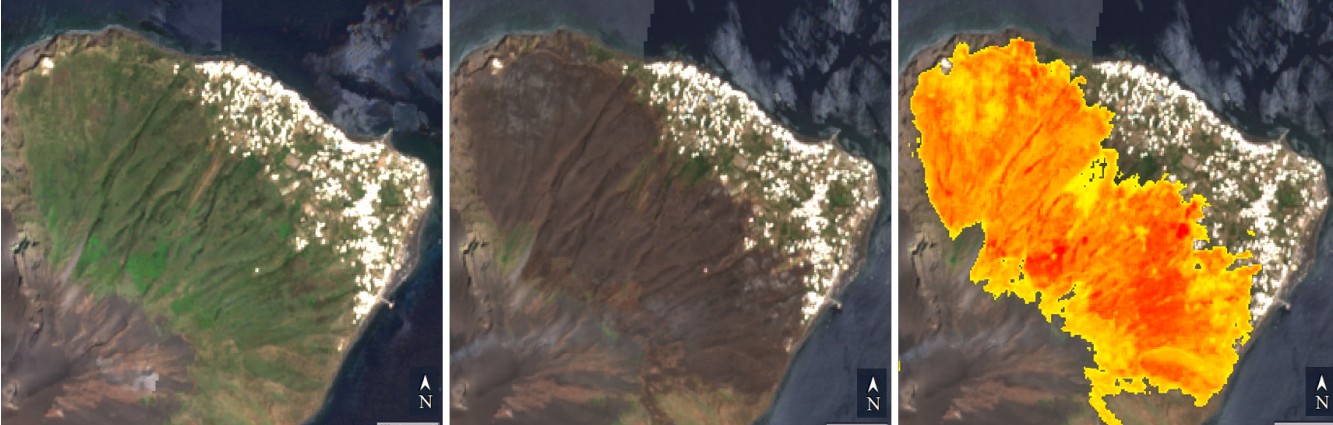

**Figure 1: Map of the study area (light green) with the place names mentioned in the text. The pink colour indicates the area where the vegetation plots for validation were sampled. Red lines identify the two transects along which the stem density of Saccharum was measured.**

**Figure 2: (from left to right) Sentinel 2 image before fire event (composite of acquisitions in the time period 22/04 - 22/05/2022); Sentinel 2 image after fire (composite of acquisitions in the time period 25/05/22-15/06/2022); *dNBR*-inferred burned area (yellow: low-, orange: middle-, red: high-severity damage) overlaid on the middle image composite.**

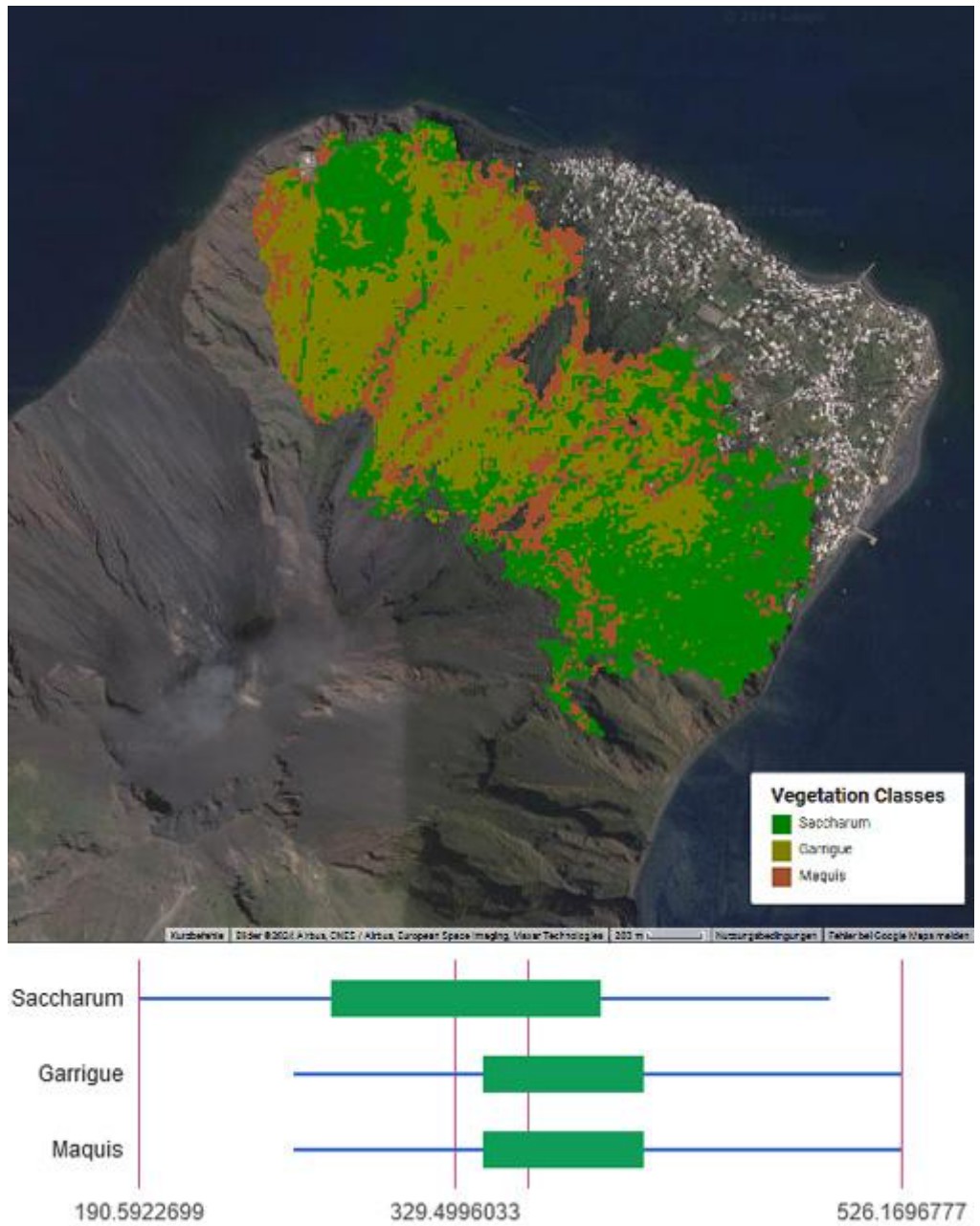

556

Figure 3: (top) supervised classification of vegetation classes in the study area, overlaid on Google Earth base map (© 2024 Airbus, CNES/Airbus, European Space Imaging, Maxar Technologies); (bottom) Boxplot showing the distribution of *dNBR* values per vegetation class, evaluated on the image composites from acquisitions in the periods 15 April - 22 May and 26 May -15 June 2022. Boxes and whiskers correspond to one and two standard deviations, accounting for 68% and 95% of the processed values, respectively. Fire occurred in garrigue and maquis was estimated to be the most severe.

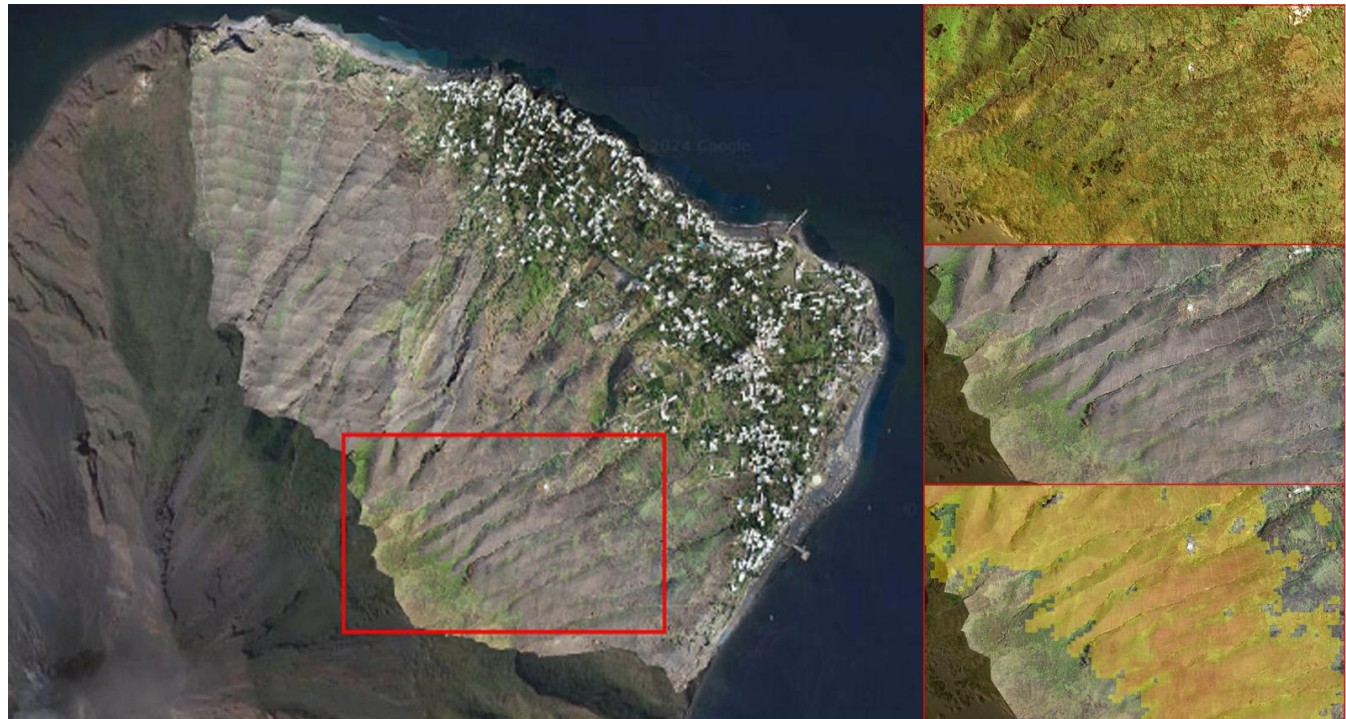

**Figure 4: (left) high resolution drone image acquired on 17 August 2022 to assess the quality of the information derived from *dNBR* analysis, overlaid on an high resolution image from Google Earth basemap; (top right) pre-fire detail from Google Earth basemap; (middle right) post-fire detail from drone image; (bottom right) same detail with overlaid thresholded *dNBR* values higher than 0.19 (using pre-fire and August 2022 scene), semitransparent for visual comparison (yellow: low-, orange: middle-, red: high- severity damage). Credits of drone images: Antonio Zimbone. Credits for Google base map: © 2024 Airbus, CNES/Airbus, European Space Imaging, Maxar Technologies.**

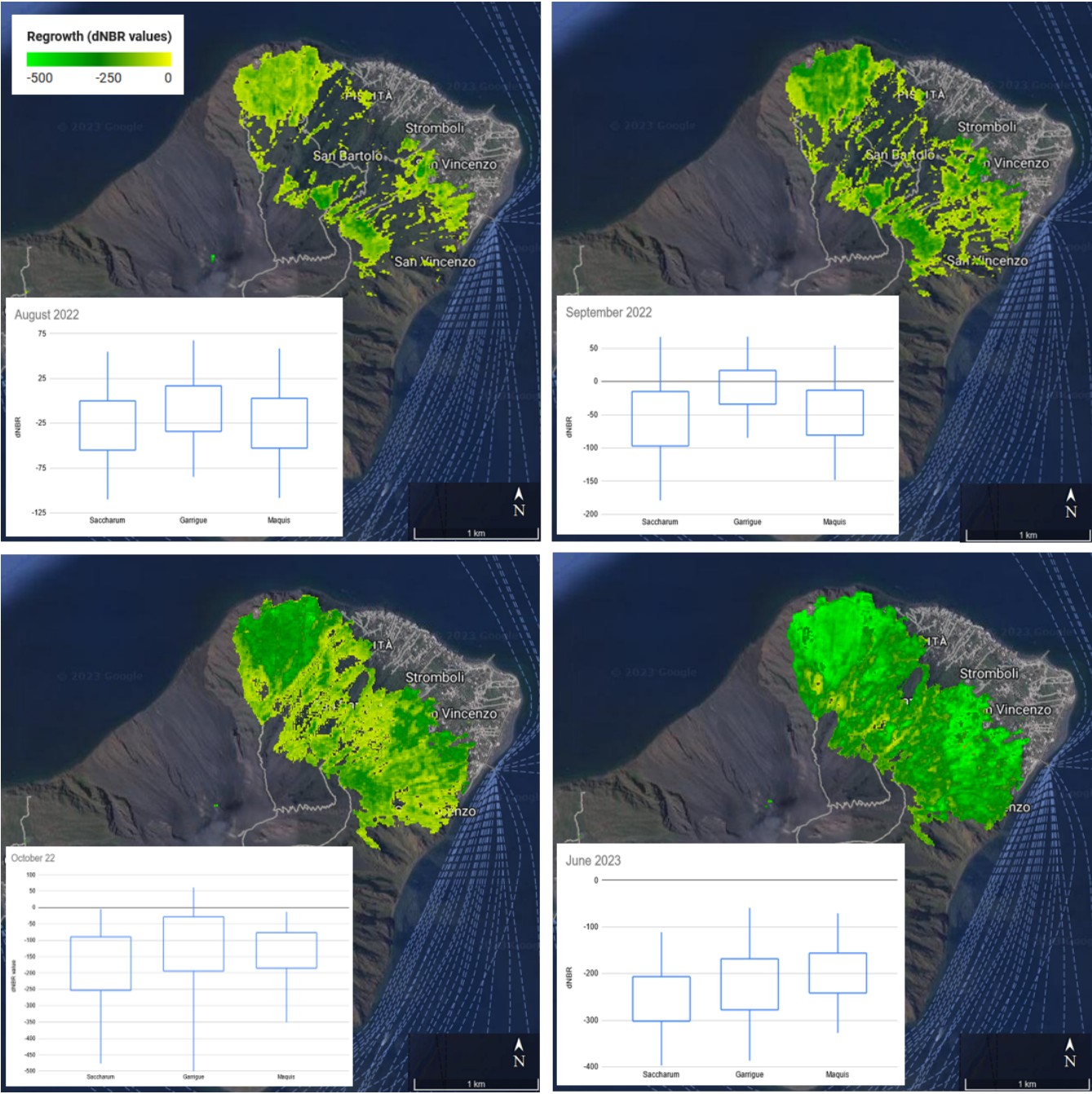

**Figure 5: Vegetation recovery in the area affected by the fire, estimated through dNBR values from different acquisitions of Sentinel-2 images, overlaid on Google Earth base map (© 2024 Airbus, CNES/Airbus, European Space Imaging, Maxar Technologies). Boxplots show the distribution of dNBR values associated with recovery in the areas occupied by Saccharum, garrigue, and maquis. Boxes and whiskers correspond to one and two standard deviations, accounting for 68% and 95% of the processed values, respectively. The following thresholds were suggested by Key and Benson (1996) to categorise levels of recovery from *dNBR* values rescaled by 1000: no change from 0 to -100, low enhanced recovery from -100 to -250, and high enhanced recovery (high) from -250. *Saccharum* is characterized by faster recovery than the maquis and the garrigue, particularly at the beginning of the first growing season after fire (September-October 2022).**

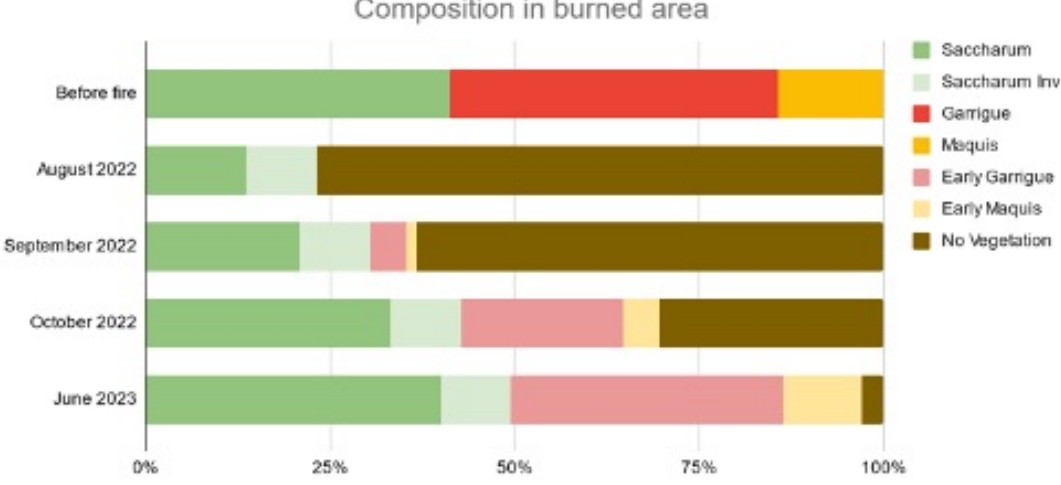

Figure 6: estimated vegetation composition in the study area (cover %). "Saccharum" vegetation patches occupied by Saccharum both before and after fire; "Saccharum Inv" sums the surface areas previously occupied by other vegetation units and invaded by Saccharum after fire. "Early garrigue" and "Early maquis" refer to early post-fire successional stages of these two vegetation classes, dominated by annual plants, resprouted shrubs and seedlings of perennial seeders, chefly *Cistus sp. pl.*

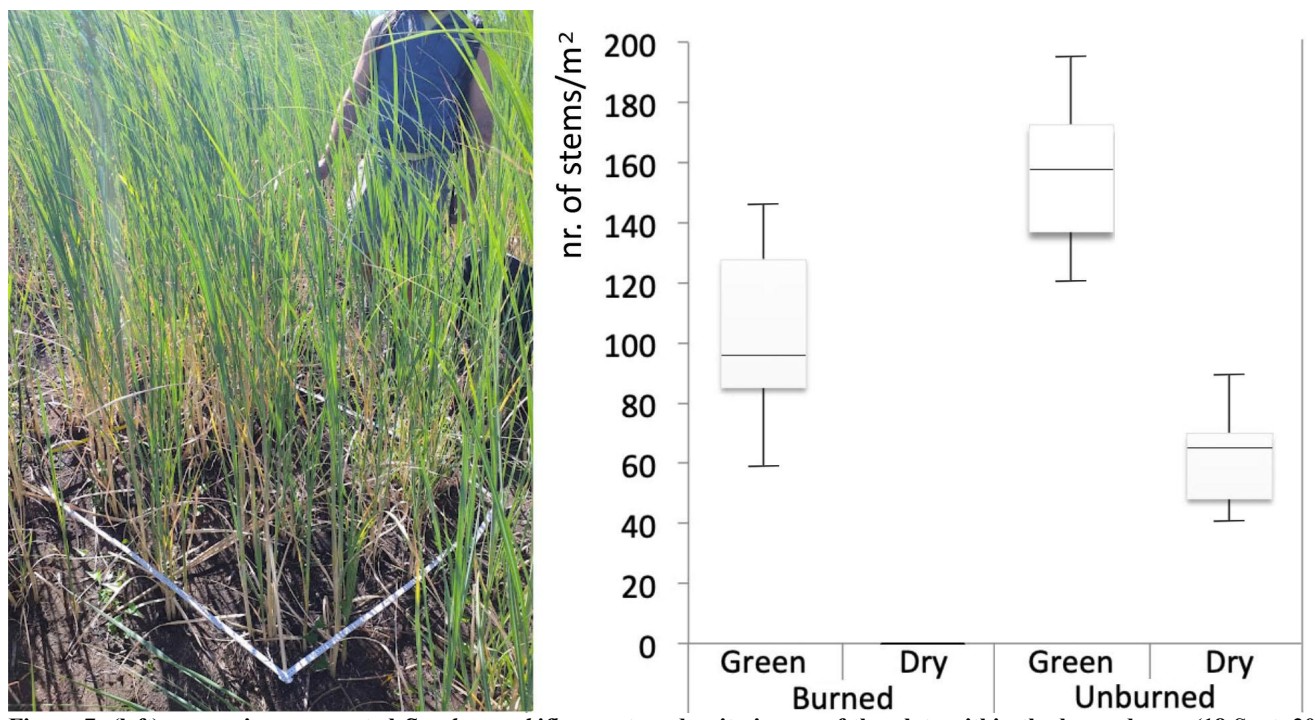

Figure 7: (left) measuring resprouted *Saccharum biflorum* stem density in one of the plots within the burned area (18 Sept. 2022, photo by R. Guarino); (right) boxplots of the stem density of *Saccharum* in burned and unburned patches.

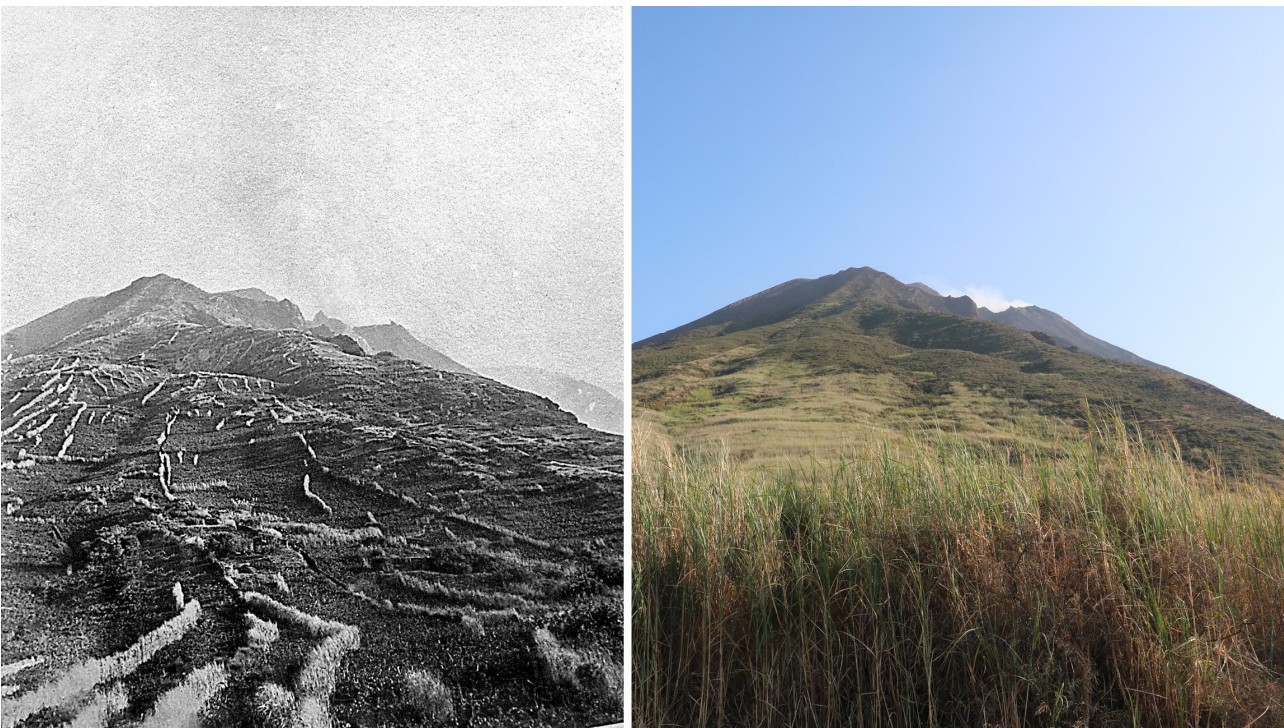

**Figure 8: (left) historical photo of terraced vineyards on Stromboli (year: 1891, anonymous), with rows of *Saccharum biflorum* used as windbreaks; (right) same view, 130 years later (16 July 2021, photo by P. Lo Cascio).**