# Peer review of "Remote sensing reveals fire-driven enhancement of a C4 invasive alien"

_Biogeosciences, 2023_

## Author Comment (AC5)

**RC1**:  Anonymous Referee #1, 13 Apr 2023

**General comments**

The papers objective is to test the power of relatively simple and straightforward indices (dNBR and NDVI) to map fire intensity, extent and the subsequent vegetation recovery of a recent fire on a Mediterranean volcanic island. The vegetation recovery seems to be dominated by an alien grass species. The title fits this objective and the topic fits the Journals Special issue: "The role of fire in the Earth system: understanding interactions with the land, atmosphere, and society". The language in the study is mainly fluent and precise, however some parts would benefit from rephrasing (specified in the "Specific comments" section). The authors use two in theory well fitting indices for their approach (NDVI and dNBR) to map the extent and to a certain degree the fire intensity. Both indices are explained and used properly. The study is able to show, that the investigated fire boosted the distribution of the invasive alien species S. biflorum, outcompeting native vegetation in the time directly after the fire.

Thank you for your careful reading of the manuscript and for fully grasping the purpose of our work. In a second revised version of the manuscript we will try to shorten the longer sentences according to the suggestions in the "Specific comments" section.

However the structure should be streamlined: parts that are now in the method section rather belong in the introduction, parts that are now in the results belong in the methods part. The mentioned ground data should be explained further. Additionally, the figures / maps need to be reworked. Overall, the study investigates an interesting event and circumstance, but lacks depth, which should be added. It is also not entirely clear to me what exactly is the focus. Is it the fact that you could successfully map fire intensity and extent plus the vegetation recovery or the fact, that S.biflorum is the one species to rapidly and successfully grow on the burned areas. In the methods/ result section, you mainly talk about the mapping, how you created your damage/recovery maps and only very briefly touch on the fact, that S. biflorum could be observed as the dominant species and did fieldwork regarding its ability to colonize the burnt area. This lead me to believe, that the mapping with Satellite imagery is the main focus. However, in the discussion you mainly talk about the S. biflorum rather than your mapping. Some general suggestions for improvement are already given, more in depth comments can be found in the next section. The abstract summarizes the content of the paper well. The Keywords include "field monitoring", however, the description of the field methods is lacking and should be talked about and explained in more detail, especially if "field monitoring" is used as a keyword.

We agree with the reviewer that some part in the method section could be put in the introduction and that the first parts of the results might best be moved in the methods section. We will do our best to streamline the structure of the paper.

As for the focus of the paper, it is indeed twofold. On the one hand, we were interested in offering the reader a case study demonstrating the superiority of the dNBR over the NDVI for identifying and quantifying fire damage; on the other, we thought it would be interesting to highlight the ecological behaviour of an invasive exotic plant in the Mediterranean and its fire-driven ability to colonise new spaces. We will try to be clearer in this sense and to better balance the contents of the sections "results" and "discussion". Also, as suggested, we will remove 'field monitoring' from the keywords, since our work does not actually focus on this issue.

**Specific comments**

*1. Introduction*:

The introduction overall manages to set the scene for the investigation. Some minor changes I would suggest:

1.1 Is the species Saccharum biflorum an alien or also an invasive alien species? In your short summary you call it invasive alien species, throughout the text you often only lable it as alien.

It is an invasive alien species. We will clarify it throughout the text

1.2 I suggest to add the part on S. biflorum (Line 167- 182) from the methods here rather than having it in the methods.

We agree

*2. Methods*:

2.1 Sensitivity of the red-edge portion of the electromagnetic spectrum to variations in vegetation cover/health has been discussed and accepted, have you tried e.g. also NDRE rather than NDVI for more precise results? Why did you exactly choose the NDVI?

[Figure]

Thanks for this remark. While designing the experiments, we considered that traditionally NDVI is more related to the extent of alive vegetation at canopy level, while NDRE is more sensitive to subtle variation in vegetation health (e.g. related to soil nitrogen concentration, clorophyll level…). Nevertheless, I applied also the NDRE and report in the attached images the differential NDVI (left) and NDRE (right) before and after the fire, after optimizing the stretch of the false color map in order for them to match. A visual assessment reveals that, indeed, the maps are comparable. We computed Pearson's correlation, which shows that the correlation with dNBR is slightly higher for NDVI:

- Pearson's correlation
  - dNBR / diff NDRE = 0.973
  - dNBR / NDVI = 0.977

Anyway, as other reviewers also point out, the analysis through NDVI will be probably have less relevance in the new version of the paper, as it is more difficult to link it to damage extent (in the literature damage classes are better defined for dNBR).

2.2 Study area section has some unneccessary information (e.g. when the first weather station was installed and when the newer one took over e.g. around lines 92 ff). However, a better overview map over the study area is missing. You reference different landmarks on the island a lot, but give no map including any labeling.

We will add a map of the study area with the place names named in the text and we will remove the unnecessary information

2.3 For the NBR you describe which bands are used to calculate it, for the NDVI this information is missing.

This is correct. We selected band 8 for NIR and band 4 for red.

2.4 Maybe include the NDVI maps in the supplements or leave out the NDVI entirely if you do not show any results on it and rather mention, that you tested it but the dNBR worked considerably better. The whole explanation icluding the formula is not needed in my opinion if you don't show any results (at least in the supplements) on it.

We agree. We will move formulas and the NDVI maps in the on-line supplements

2.5 Line 167- 182 rather belong in introduction than in methods. By this time, it should be established, why the species S.biflorum is of interest.

Thanks for your suggestion. We will modify the text accordingly.

2.6 For me, a map of the location of the carried out fieldwork is missing. Overall, the part about the evaluation of stem density (Line 182-187) was not clear for me: did you collect

this data? If you collected the data for this study, I would expect more information on the field sampling (transects? plots? distance between the unburned/burned patches which are compared? How will you compare them? When did you sample it? Location?). You mention the time of field sampling only in the description of figure 3.

We will add methodological details on the fieldwork and the location of the sampling sites

Overall, the method part would benefit from a table including what data was used (which satellite images from when/drone images/field data?), what analyses where done with it (dNBR, NDVI…) and for what porpuse (map fire intensity/extent/recovery..). Right now, there is not all the data mentioned in the method section for which results are showin/described. Additionally, more information on the fieldwork and the location of the sampling sites would be beneficial.

We will add a table (or a flow chart) in the methods part

*3. Results*

3.1 Line 189 f: The part about the fixed threshold should be moved to the methods and explained better. Why exactly this threshold? What made you decide on it ? References? Your own assessment?

We agree this should have been better clarified. We actually started from the widely used definition of burns severity classes from Keeley (2009): https://pubs.usgs.gov/publication/70032718

We used then the following reference to obtain our first map with hard classes (not reported in the paper)

| Severity Level | dNBR Range (scaled by $10^3$) | dNBR Range (not scaled) |
|---|---|---|
| Enhanced Regrowth, high (post-fire) | -500 to -251 | -0.500 to -0.251 |
| Enhanced Regrowth, low (post-fire) | -250 to -101 | -0.250 to -0.101 |
| Unburned | -100 to +99 | -0.100 to +0.99 |
| Low Severity | +100 to +269 | +0.100 to +0.269 |
| Moderate-low Severity | +270 to +439 | +0.270 to +0.439 |
| Miderate-high Severity | +440 to +659 | +0.440 to +0.659 |
| High Severity | +660 to +1300 | +0.660 to +1.300 |

[Figure]

As it can be seen, false alarms appear in the urban area and in the west side of the island. As the division in classes of damage should be adapted to the case at hand, we considered that a conservative threshold should be applied, in order to identify an area which was damaged for certain. The value in the middle of the "Low Severity" class and rounded to the second decimal digit, 0.19, was selected as the lowest value without visible false alarms in the results.

3.2 Line 200-206: Rather belongs in methods

We agree. We will move it there

3.3 Line 209 f: You describe using Satellite images from August 15th-17th to calculate a new dNBR- those images are not mentioned before in the methods.

We will provide these details in the methods

3.4 Line 227f : You describe the usage of Satellite images from September 2022 (which again are not mentioned before in the methods)

We will provide these details in the methods

3.5 Line 232: How did you verify the patches being S.biflorum? Visually?

We combined spectral fingerprinting with field surveys and photointerpretation

Overall, the results seem to be a little shallow. I would expect some more analyses investigating a potential relationship between fire intensity and recovery for example. Also the vegetation before the fire compared with the vegetation after the fire (based on NDVI / NDRE maps) could be interesting in terms of vegetation recovery. Additionally, the ground data mentioned in the methods are only plotted, but not analysed/no analyses shown (testing for significant differences eg?). It is not clear to me, to which extent the field work is connected to your remote sensing analyses.

We will test the homogeneity of variance and the correlation between vegetation types and the fire intensity. As for the vegetation recovery, in order to provide a more circumstanced response, we went to Stromboli in the first half of September 2023 to make additional surveys in the study area. We are now checking whether the data collected have enough resolution to implement the paper with an analysis of the variation in the area of occupancy of the native vegetation and of the target species (Saccharum spontaneum) one year after the previous survey. This would substantially improve the content of the paper and make it much more focused.

Additionally, in the results are often parts mixed in that belong in the methods. Reread and streamline the part with that in mind, that in the results section only the results derived from analyses described in the methods should be presented.

Ok, we will do so. Thanks for noticing

*4. Discussion*

4.1 Lines 234-240:  Move explaining/justifying your method to another part of the discussion. Start with discussing your results in context of other literature.

Ok, we will do so

The discussion focuses on the role of S. biflorum on vegetation recovery. However, the results focus on mapping fire intensity and extent and map recovery. Your field data is simply plotted, no analyses for this data is shown and the field data simply provides proof, that S. biflorum grows quickly after fire in the affected areas. Apart from the short part in the beginning, you don't talk about your methods for mapping anymore. I would expect some comparison to other literature/studies also mapping fire/fire intensities (e.g. Weiser et al. (2021) (also on an oceanic island) or Gibson et al. (2020)) additionally to the in depth discussion about the recovery dominated by S. biflorum.

We will add a comparison to other studies in the discussion

5. *Conclusion*

5.1 Line 293 -298 : You introduce new information, which maybe fits better in the discussion. In the conclusion I expect to find the main takeaway message from the study, not an entirely new idea (S. biflorum being desirable to provide erosion protection).

We will modify the conclusion accordingly

**Figures:**

*Figure* 1

The maps need to be reworked. Basic necessities for maps are missing (no direction (north arrow), no legend, no scale etc). Additionally, there is too much unnecessary ocean/land mass that is of no interest.

Suggestion: Include an overview map to show the location of the island, and than zoom in and show only the parts of interest (the northeastern part of the island). As those map are the heart of your study right now, they should be much more informative and better.

We will modify Figure 1 according to your suggestions

*Figure 2*

Again, no north arrow, no scale, no legend. Additionally, to be able to compare, the zoomed in drone image without the overlapping dNBR results added would make the reader able to really compare the damage map with the drone image.

We will modify Figure 2 according to your suggestions

**Technical corrections**

Line 64: unprecise phrasing. It sounds like the Island of Stromboli is the volcano, but as I understand, the island is called "Stromboli" and the volcano on it is Mount Stromboli.

The Island of Stromboli is actually the summit of a volcano that rises from a seabed at a depth of about 2000 m. The Island is 925 m a.s.l. and consists of a single cone. So, it is not so common to distinguish between Stromboli and Mount Stromboli

Line 102: mentioning of holm-oak, but missing the scientific name (quercus ilex)

We will add the scientific name

Line 147 f: Rephrase, unprecise phrasing, verify, if "whose" is the right word to use here.

Ok

150 f: not good English, rephrase

Ok

Line 151: capitalize the first letters for Normalised Difference Vegetation Index (as you did for the NBR).

Ok

Line 191: you talk about, how close the fire gets to the inhabited area according to figure 1. However, Figure 2 would be more of a fitting example, as it is easier to see.

Ok

Line 484: spaces are missing in-between words

Ok

**References mentioned:**

Gibson, R., Danaher, T., Hehir, W., & Collins, L. (2020). A remote sensing approach to mapping fire severity in south-eastern Australia using sentinel 2 and random forest. Remote Sensing of Environment, 240, 111702.

Weiser, F., Sauer, A., Gettueva, D., Field, R., Irl, S. D., Vetaas, O., ... & Beierkuhnlein, C. (2021). Impacts of forest fire on understory species diversity in Canary pine ecosystems on the island of La Palma. Forests, 12(12), 163

...Thank you so much, again, for the thorough review!

---

## Author Comment (AC6)

**CC1**: 'Comment on bg-2023-19', Gianluigi Ottaviani, 03 Aug 2023

The manuscript entitled "***Remote sensing reveals fire-driven facilitation of a C4 rhizomatous alien grass on a small Mediterranean volcanic island***" by Guarino et al. presents an interesting case study about the short-term impacts of, and changes caused by a fire event on the vegetation of the Mediterranean island of Stromboli (Aeolian archipelago); in 2022, ~50% of the vegetated area was affected by fire, with variable intensity.

I consider the research topic relevant to the readership of the Journal and of the Special Issue. The authors used remote sensing methods and indices suitable to capture the short-term changes associated with the fire (pre- and post-event).

Thank you for carefully reading the manuscript and for all your suggestions and observations.

However, I remain quite skeptical about the way the authors link to the dynamics (especially in relation to possible future scenarios) of the invasive species *Saccharum biflorum*. I am afraid that I do not share their positive view about a possible/desirable decline of this species on Stromboli over time. *S. biflorum* is extremely competitive thanks to a variety of functional strategies (e.g. large resource allocation belowground into clonal and bud-bearing organs which can boost a quick resprouting and local spread/space occupancy/resource uptake) under current and probably also under predicted conditions (likely more disturbed) which could affect and define different ecosystems on Stromboli. Along this line, I would be more careful.

Thank you, we will try to be more nuanced on this point (but see Richter and Lingenhöhl, 2002). You are right, Saccharum behaves like a very competitive species on Stromboli. But the fire frequency could make the difference. We cannot ignore the decreased Saccharum area of occupancy shown in the vegetation maps of Stromboli published by Richter and Lingenhöhl (2002, Fig. 4), comparing the vegetation cover in 1984 and in 2002, after 18 years of no destructive fires. This is in agreement, for example, with the demographic boom shown by Cytisus aeolicus during the last decades (Zaia et al., 2020).

Also, if I am not mistaken, there is no reference/inference related to the analysis and results showed in Fig. 3 reporting the variability of *S. biflorum* stem density under different field experimental treatments, which can be instead interesting to discuss (alternatively, this analysis and associated results could be removed from the story, if the authors consider them marginal).

We will add some sentences in the discussion about Saccharum's ability to resprout immediately after fire, with a density of green stems only slightly lower than that of unburnt patches, thanks to the large resource allocation belowground.

All analyses related to fire-induced changes are interesting, yet quite descriptive without any statistical testing – quite an unusual decision for a regular research article. At the same time, I tend to agree with the authors when they acknowledge that *S. biflorum* may also play positive functional roles, such as in slope and soil stabilization against erosion.

We will test the homogeneity of variance and the correlation between vegetation types and the fire intensity.

At last, the tone of the language used is sometimes too colloquial, and I encourage the authors to consistently use a more formal language throughout (I spotted a few cases, suggesting alternatives below in the specific/line numbered comments).

we will do our best to use more formal language

Specific comments

L1: in the title, the reference to facilitation may be misleading/misplaced in the context of this paper. This term comes with a clear connotation in community ecology, e.g. facilitation of cushion-like plants tending to facilitate the germination and persistence of herb species in alpine environments. I would therefore avoid this term throughout.

Ok, we will replace this term with "regeneration"

L25: replace "arson" with "fire"

Ok

L28: "recurrent" – is fire really recurring with a predictable regime on Stromboli?

Not really predictable… We will replace this term with "repeated"

L30: "complex" - I am not a big fan of using this term (i.e. we tend to use this adjective when we are not able to disentangle and grasp what is behind a pattern), yet all is complex in ecology! Here, for example, using "multifaceted" or "dynamic" would do

Ok

L32-33: is this a hypothetical trajectory or you have reference/evidence to back this claim up? Of course, this could be eventually elaborated in the Discussion and Conclusion (but see my related comment there)

As we wrote in the text, there are reasonable indications that, if the vegetation is not affected by fire too frequently, Saccharum could be gradually be replaced by native vegetation within a few decades, as captured in the maps published by Richter and Lingenhöhl (2002, Fig. 4).

L33-35: as for the previous sentence – is this your inference/speculation backed up by your data or contextualized with other studies? It may or may not be the case. Also, consider replacing "beds" with "patches" throughout

We have intentionally used the conditional, but the possibility of a relatively fast decline of Saccarum if local vegetation is no more affected by frequent fire is not pure speculation. As we wrote in the text, this has been already postulated by Ferro & Furnari (1968), and captured by the maps published by Richter and Lingenhöhl (2002, Fig. 4). The use of "beds" is commonly used to describe the layer formed by tall grasses (see, for instance, "reedbeds")

L39: "frequency" – not only that, but also intensity/severity which together shape the fire regime in an area

Ok

L43: "functional diversity" – of what? Plants, animal, soil microbiome, etc?

Of the soil mocrobiome

L45: "human habitat" – replace with "open-canopy space available for human activities"? Human habitat seems a bit too strong to me. In the same line, "foraging"; this relates to both humans and animals as well, right?

Ok

L48: "scrub" – while I acknowledge that this is a stylistic choice/taste, I would prefer "shrub" and "shrubland" - this is the terminology used in many biome-focused global studies, and relates directly to the main growth form defining Med-ecosystems. Also, maybe better stating "woody plant" as not only shrubs may have been advantaged by abandonment?

Ok, we will replace "scrub" with "woody plant" (L48) and with "shrub" elsewhere else

L53: "Climate change scenarios indicate rising temperatures and decreasing amounts of precipitation" – this is right, and may cascade to lower aboveground biomass production, which may therefore limit fuel availability (while being drier and easier to ignite). Such types of feedbacks, sometimes with opposing effects, are increasingly addressed by climate-change modelers (e.g. Baudena et al. 2020 New Phytol for a study on post fire-aridity interactions) and may play a key role in shaping vegetation dynamics on Stromboli as well

Thanks for the suggestion. We will implement it in the text.

L56: "shorter" – not only shorter fire intervals but potentially also more intense fires? Also, in the same line, I would say "potentially followed by alien" instead of "alien" only

Ok

L58: "unidirectional change" – not sure I am grasping what "unidirectional" may mean/imply here; please, specify

We specified that unidirectional change in invaded ecosystems means that invasive species able to sustain an increased fire frequency and intensity may generate favourable conditions for their self-perpetuation.

L62-63: hence, especially exposed to a variety of disturbances, including fires induced by eruptions?

Exactly

L65-66: i.e. following classic predictions of the Theory of Island Biogeography (as for the seminal work by MacArthur & Wilson 1963, 1967)

This is explicitly mentioned in the reference mentioned at the end of the sentence (Chiarucci et al., 2021)

L69: "volcanic ash" – this being a source of nutrients which this alien species can exploit more readily than native species (because strong competitive [C-strategy according to Grime's CSR model])? I put it out here, yet it may be handy when doing inferences

Thank you for this suggestion. As a matter of fact, it would be interesting to test whether the clear preference of Saccharum for thick deposits of volcanic ash depends on nutrient availability, on reduced mechanical resistance to the fast eloncation of rhizomes or on a combination of both.

L70: "arson" – being caused by volcanic activity, this is not an arson (man-made). I would also replace or specify in the following wording "paroxysmal activity"; perhaps, better saying "volcanic eruption" as more directly related to/describing the natural phenomenon

Ok

L76: "quadrants" – "sides" or "slopes"?

Ok

L77-80: the scope of this work is fine and clear, however, the authors may frame the aim of this study as e.g. 2 questions with related hypotheses instead of keeping it fully "explorative"?

Thank you for this suggestion. We will substantially rewrite the final part of the introduction also following the suggestions of Reviewer 1

L84: I would say "maximum elevation"

Ok

L107: "due to recklessness during the filming of a television drama" – I would simply refer to a human-generated fire (or arson directly, which carries this connotation)

Sorry, but we believe that it is interesting to give a little detail on how such a large fire begun.

L109: delete "by chance"

Ok

L139: "some soil conditions" – meaning, bare rock outcropping/scorched after fire?

According to Lentin et al. these include charred organic material (litter, duff, and dead wood), bare mineral soil, and ash.

L180: delete "somewhat"

Ok

L183-184: when & where (e.g. plots were randomly distributed)?

We will add methodological details on the fieldwork and the location of the sampling sites

L185: "in order to" – this formulation is used very often but simply using "to" can suffice in most cases

Ok

After L187 before Results: related to one of my general comments/concerns; there is not a section dedicated to "Data/Statistical analysis"

We will add it.

L191: "surrounded the Osservatorio Restaurant" – what is the relevance of specifying the name of a restaurant? As abovementioned, such level of colloquialism should be avoided

Ok

L192: "NDVI values were strongly correlated with dNBR values" – to what extent? Any values? Strong correlation could also have deep implications when interpreting results and performances of these parameters (even though the following sentences explain differences between and sensitivities of these two parameters)

We agree that this is relevant: the correlation between differential NDVI values and dNBR was very high: 0.977. As reported, dNBR values appeared more consistent, possibly also

due to the inclusion of a spectral band at 20 m which prevented some false alarms and noisy results.

L204: what is the new threshold (previously set > 0.19)? Is this threshold standard in such studies?

We agree this should have been better clarified. We actually started from the widely used definition of burns severity classes from Keeley (2009): https://pubs.usgs.gov/publication/70032718

We used then the following reference to obtain our first map with hard classes (not reported in the paper)

| Severity Level | dNBR Range (scaled by $10^3$) | dNBR Range (not scaled) |
|---|---|---|
| Enhanced Regrowth, high (post-fire) | -500 to -251 | -0.500 to -0.251 |
| Enhanced Regrowth, low (post-fire) | -250 to -101 | -0.250 to -0.101 |
| Unburned | -100 to +99 | -0.100 to +0.99 |
| Low Severity | +100 to +269 | +0.100 to +0.269 |
| Moderate-low Severity | +270 to +439 | +0.270 to +0.439 |
| Miderate-high Severity | +440 to +659 | +0.440 to +0.659 |
| High Severity | +660 to +1300 | +0.660 to +1.300 |

[Figure]

As it can be seen, false alarms appear in the urban area and in the west side of the island. As the division in classes of damage should be adapted to the case at hand, we considered that a conservative threshold should be applied, in order to identify an area which was damaged for certain. The value in the middle of the "Low Severity" class and rounded to the second decimal digit, 0.19, was selected as the lowest value without visible false alarms in the results.

L205: "pixels"

Ok

L207-208: "Of these, 44.31 ha showed high severity burning, assigned to a dNBR value higher than 0.45" – is this threshold value standard?

This is the threshold fixed by the European Forest Fire Information Service (EFFIS, 2022). We will mention the reference in the text.

L214: "strong vegetative stress" – what this means/implies?

It implies in this case suffering extreme changes in temperature, possibly humidity and water availability in the area, aspects that can affect the plant's metabolism and development.

L215: "fire event"

Ok

L223-225: This sentence could be streamlined as: "The fast recovery of *Saccharum* patches was evident by the stark contrast between the green colour of *Saccharum* and the surrounding black/burned landscape - further emphasised by the particularly hot and dry summer of 2022." Also, please avoid colloquialism ("caught everyone's attention")

Ok

L229-231: this is related to the field-based plots, comparing burned vs. unburned areas; yet, no sum stats of the analyses are reported

We will add some stats

L237: consider rephrasing "to its ground being exposed" as "bare ground exposure"

Ok

L242-244: how can this be related to degradation if recurrent fire is an eco-evolutionary force shaping Med-ecosystems for My (as mentioned in previous part of the sentence)? If the authors are willing to maintain this point, they should elaborate more in depth this nuanced task

Ok

L244-248: surely certain species could be largely and detrimentally affected by fire in the short-term (this study was executed right after fire). However, one should consider the time-scale involved in post-fire regeneration, and fire regime has been greatly altered (e.g. by fire suppression and no prescribed burning as done e.g. in Australia, US). Many Med-species are fire-adapted and require fire with a given regime (in terms of frequency and intensity/severity). Unfortunately, I do not see how the authors can back up their claim –

please, refer to the extensive work by e.g. Pausas, Lamont, Keeley, Bond (among many others) over the last 2-3 decades. Also, while fire on islands can be considered less likely to occur due to milder and more humid conditions associated with a climate buffering effect of sea/ocean than on mainland counterparts (e.g. Burns 2019 book on the Island syndrome), on active volcanic islands fire ignition can occur during volcanic eruptions. Hence, this assumption does not necessary hold here and one could also see a case for the opposite reasoning

Thanks for these suggestions. We will duly consider them in a second, revised version.

L249-264: I like these two sections (L249-263); in my opinion, these do a much better and more balanced job than the previous one to contextualize the results in a sound eco-evolutionary frame

Thank you

L272-277: I struggle to follow the reasoning of this section, and cannot see how the authors can back up their claim. Fire was, is and will be on Stromboli (i.e. complete fire suppression could be even more impacting over mid- and long-terms in the face of the ongoing and exacerbating climate change/warming towards more extreme events), so how can the vegetation (also shaped by a given fire regime) not being affected by fire? I am afraid that the damage caused by the introduction of such a successful invasive species (equipped with many functional strategies making it super-competitive and responding well to fire and other environmental factors) will be long-lasting. I am therefore sorry not to share the authors' positive view on this predictive point

You are right, fire was, is and will be on Stromboli. But the fire frequency could make the difference. Some hints on the speed of the vegetation dynamics are offered by the decreased Saccharum area of occupancy shown in the vegetation maps published by Richter and Lingenhöhl (2002, Fig. 4), comparing the vegetation cover in 1984 and in 2002, after 18 years of no destructive fires. This is in agreement, for example, with the demographic boom shown by Cytisus aeolicus during the last decades (Zaia et al., 2020).

L287-288: also, being a very good resprouter from rhizomes, *Saccharum* may be even favored by herbivory

Ok

L289: Conclusions – I like the cautious way the authors framed this concluding paragraph, interpreting in a balanced manner detrimental vs positive effects at various levels associated with the presence and spread of *Saccharum* on Stromboli

Thank you

L292: "pioneer role" – for the reasons exposed above, *Saccharum* behaves way more than a pioneer (e.g. an engineer species as exposed further below), and is there to stay unless

time-consuming, long, expensive management practices (such as complete removal of above- and belowground plant parts throughout the island, quite unrealistic) are put in place

Ok, we will consider this

L303: sowing not only native woody species, I guess

You are right.

L304: more specifically, how the proposed rewilding should look like? This term/notion is still hotly debated in the scientific community, and if the authors want to maintain this ground, they should explain this point in more detail

Ok, we will do so.

L487-488: Fig. 3 – as mentioned above, where are these results discussed?
We will add some sentences in the discussion about Saccharum's ability to resprout immediately after fire, with a density of green stems only slightly lower than that of unburnt patches, thanks to the large resource allocation belowground.

**Citation**: https://doi.org/10.5194/bg-2023-19-CC1

---

## Author Comment (AC7)

**RC2**: , Anonymous Referee #2, 28 Aug 2023

The study addresses a single fire event in a small Mediterranean island, estimating the area burned and its distribution by vegetation type and providing fire severity maps (but the authors don't even give figures for each severity class extent). Then some account of the short-term response of the invasive grass is reported. I don't think the results are relevant enough to warrant publication as a full-fledged paper and I would see it better as a research note (if included within the journal's types of papers). Also, the authors do not seem familiar enough with fire terminology and concepts, see comments below.

Thanks for your review. In order to make the contents more relevant to an international readership, we will test the homogeneity of variance and the correlation between vegetation types and the fire intensity. As for the vegetation recovery, in order to provide more circumstanced results, we went to Stromboli in the first half of September 2023 to carry out some additional surveys in the study area. We hope this would substantially improve the content of the paper and make it much more focused. We will revise the fire terminology and concepts following your comments below.

Specific comments

L48. Shrub and shrubland should be used preferentially to scrub and scrubland.

We will follow your suggestion, thank you

L51. Arson respects to incendiarism, not to negligent fires. Correct here and elsewhere.

We will delete "due to arson,"

L77. Postfire damage?

We will replace "the post fire damage on local vegetation" "the effects of fire and the short-term vegetation response"

L82. I don't think the Study area description should be so exhaustive/detailed. Please revise and maintain only what is relevant for the reader to understand the study context.

We will shorten the study area description, also following the suggestions of Reviewer 1

L109. By chance or by design, given differences in fuel characteristics? You cannot really know, so please eliminate "by chance".

Ok

L111. You are assessing degree of change. Damage can be hypothesized or inferred, but it is not being measured. Change here and elsewhere.

We will replace it with "fire severity", "fire-driven vegetation loss", "fire-driven vegetation changes" or "fire-driven vegetation dynamics", depending on the context

L142. "the best performance". This is debatable, as several studies have shown that other NBR-based indices provide a better assessment of fire severity.

Agree, we will replace "has the best performance" with "performs well"

L146. Replace "damage severity" by "fire severity". A positive dNBR indicates biomass change (consumption or scorch).

Ok

L167. "bushy grass" seems awkward, please improve.

"bush grass", maybe?

L167-187. Again, all this description is unnecessarily long.

We will shorten it

L190. A severity map.

Ok

L200. Revise the Results to move methodological components to Methods.

Ok

L235. You can write "burned" or equivalent, but don't use "destroyed" as it conveys a charged and potentiallybiased perspective of fire effects.

Ok

L289. The Conclusions are not really conclusive, as new information is introduced and discussed. Perhaps the study does not even need a Conclusion, and concluding remarks can be included in the Discussion.
We will modify the conclusion accordingly

**Citation**: https://doi.org/10.5194/bg-2023-19-RC2

---

## Author Comment (AC8)

**RC3**: 'Comment on bg-2023-19', Anonymous Referee #3, 06 Sep 2023

The researchers studied the extent of a fire on the island of Stromboli and its effects on the vegetation cover, as well as its recovery using NDVI and dNBR. They conclude that, although half of the vegetation of the island was affected, the alien species Saccharum biflorum experienced a fast recovery and dominated the area, occupying also the patches with former native vegetation.

Overall, I think it is an interesting topic, relevant to address, but, in my opinion, the researchers could go further in the study, it lacks challenges. The study shows the vegetation cover affected and the great capacity of Saccharum biflorum for recovering. However, it does not contribute with deeper ecological knowledge or new methodology. There is a lack of statistical analyses (e.g., differences on stem density), there is no data about the area covered by native vegetation before the fire and after the fire. Although in the abstract the researchers introduce the intensity as a study variable, it is not addressed. Alteration of the ecosystem functioning and structure is mentioned but there is no discussion about how they are altered. The writing style is too colloquial in some parts. In the methods section, there is no need to explain so much about the basic remote sensing techniques (blue, green, red channels, ...). In addition, there are some parts of results that should go in methods.

The researchers talk about the effect of more frequent fires, but they do not give data about fire frequency in this area. They also affirm that local vegetation outcompetes in the long term the alien species, but this is bases on a photography without being analysed. I would suggest trying to quantify the covers in the photography, or search for orthophotos, if they exist and try to collect photos at different years to see how the replacement evolves.

Many thanks for your careful reading of the manuscript. We will address all of your suggestions. In particular, we will test the homogeneity of variance and the correlation between vegetation types and the fire intensity and we will check the significance of the differences on stem density (Tukey's HSD, $P < 0.05$). As for the vegetation recovery, in order to provide more circumstanced results, we went to Stromboli in the first half of September 2023 to make additional surveys in the study area. We are now checking whether the data collected have enough resolution to implement the paper with an analysis of the variation in the area of occupancy of the main vegetation units in the burned area (Maquis, Garrigue, Saccharum) before and after fire, as well one year after the previous survey. This would substantially improve the content of the paper and make it much more focused.

I think the manuscript needs more work. At the moment, is very descriptive and more interesting information can be extracted.

I show my specific comments in the following lines.

Title

I would not use the noun "facilitation", but "favouring" or something like that.

Ok, we will replace "facilitation" with "regeneration"

Abstract:

- Line 17: I think the intensity of the fire is not addressed in the study.

Ok, not intensity, but severity based on dNBR thresholds proposed by the European Forest Fire Information Service (EFFIS, 2022). We will mention the reference in the text and test the correlation between vegetation types and fire intensity.

- Line 25: arson is too specific as it refers to an intentional fire. The species benefits of fire, independent of intentional or natural.

Ok, we will replace "arson" with "fire"

- Line 27: "few months": be more specific please.

Ok

- Line 28: "recurrent fires" Could you give information in the section of study area about the fire frequency of the area?

No, unfortunately we do not have these records

- Line 29: the impact in the structure and function is not studied. No data is shown about the plant communities in control areas or before the fire. After fire, the area that was with the invasive species is again with the invasive species. No data about functioning is shown.

Ok, we will be more nuanced on this point. However, we do have data on the vegetation cover before the fire and we will add a vegetation map. You are right, after fire, the area that was with the invasive species is again with the invasive species, but after one year the area of occupancy enlarged. As written above, we went to Stromboli in the first half of September 2023 to carry out additional field surveys in the study area, so to provide additional data on this point.

There is no explanation about the natural succession progress if the alien species would not be there.

We will add some sentences on this topic in the discussion, even if it is out of the scopes of the paper

We do not know when agriculture was abandoned and if patches with this alien species have been replaced by natural vegetation, after how much time and the characteristics of these areas.

It is hard to get precise records on the abandonment of agriculture, but as far as Saccharum is concerned, there is a strong indication that its area of occupancy could be largely dependent on the fire frequency. Two vegetation maps of Stromboli, published by Richter and Lingenhöhl (2002, Fig. 4), compare the vegetation cover in 1984 and in 2002, after 18 years of no destructive fires, showing a clear reduction in the areas occupied by Saccharum, in favour of native vegetation (garrigue and maquis).

- Line 34: try to measure the evidence that this alien species is replaced by natural vegetation with time. After how much time?

Within a few decades, as we wrote. Unfortunately, we cannot be more precise on this point.

Introduction

- Line 53: specify the characteristics of the areas that are affected by climate change in this way.

Ok

- Line 56: "changes in fire regime resulting in shorter fire intervals". This depends on the place. There can be fewer fires, biomass accumulation followed by very big fires but less frequent. Is the first one the case for Stromboli?

We don't know. The fire frequency in Stromboli is variable and we do not have precise records on it.

- Line 60: I would start the new paragraph about islands and Stromboli from "Small islands …"

Ok

- Line 67: I would specify that this alien species is invasive.

Ok

- Line 72: "somehow" instead of "somewhat"? Do you have any idea which factors could have enhanced the development of native scrub?

Ok, "somehow". Our idea is the lack of human disturbance and a decreased fire frequency, as suggested by the study published by Richter and Lingenhöhl (2002), mentioned above.

The areas that were burnt, were cover only by Saccharum biflorum?

No. We will try to be clearer on this point

I think specific questions, objectives or hypothesis are missing.

Thank you for this suggestion. We will substantially rewrite the final part of the introduction also following the suggestions of Reviewer 1

Methods

- Line 91: What do you mean by "smooth texture"?

It refers to the geomorphological concept of surface roughness.

- Line 102: how much vegetation cover correspond to Saccharum biflorum?

We will quantify this and add a pre-fire vegetation map

- Line 114-122: It is not necessary to explain all the details about the images of Sentinel-2. I would give only the most relevant information: Sentinel-2, resolution, …

Ok, we will shorten this part

- Line 131-134: too much explanation about the bands used for real colour images

Ok, we will shorten this part

- Line 137: no need to specify the bands corresponding to SWIR and NIR in Sentinel-2, only the equation of the index.

Ok, we will shorten this part

- Line 141: "… estimate biomass loss": do you estimate this? There are no results for this variable.

No, we don't, we will replace "biomass loss" with "vegetation loss"

- Line 146: Is it possible to differentiate through the signal the recovery of Saccharum biflorum and the native flora?

We are preparing a new version of the manuscript in which the images are classified according to vegetation units (Saccharum-dominated patches, garrigue, maquis), in order to detect Saccharum in the different stages (before the fire, right after the fire, and at different stages of regrowth).

- Line 175:  I would remove all the paraphrase and focus on the fact that Ferro and Furnari reported the species at those locations.

We prefer to report the sentence by Ferro and Furnari, because it sets the Saccharum ecological behavior so well.

- Line 180: Is it not possible to measure the cover in the photos or, are there orthophotos available to be more precise?

This would be out of the scopes of our study

I would change "somewhat" for "somehow"

Ok

- Line 182: I would explain, if there is information, how this species is favoured by fire

Ok

- Line 183: it would be good to have a map showing the sampling plots and also to specify the plot size, minimum distance between them and the design (randomly distributed, following by a pattern, ...). Also to specify if they are in similar environmental conditions.

Thank you. We will add methodological details on the fieldwork and the location of the sampling sites

Statistical analysis is also necessary to determine the differences between number of stems and percentage of dry stems between burned and unburned areas.

We will run a Tukey's post hoc test and integrate the results in figure 3.

It would also be important to assess the area of native vegetation lost and replaced by this alien species, and if the stem density is similar in the areas occupied previously by S. biflorum vs. areas occupied by natural vegetation.

We will do it

Results

-Line 192: NDVI values strongly correlated with dNBR values. Which correlation did you use? Could you show the graphic of the correlation or the parameters of the correlation?

We will put the results of NDVI and the graphic of the correlation in an on-line supplement, as suggested by Reviewer 1

- What is the advantage of using both indexes?

No advantage. We were interested in offering the reader a case study demonstrating the superiority of the dNBR over the NDVI for identifying and quantifying fire damage

- Line 215: I would not speak about biomass loss but cover loss.

We agree (and modify the text accordingly)

- Line 220: could you give an area or percentage of the native vegetation patches affected? How did you get this information, with the Sentinel image pre-event?

We will add data on the vegetation cover before the fire and a vegetation map of the burned area.

- Line 223: This sentence is too colloquial

We will rephrase it as follows (as suggested by Reviewer 2): "The fast recovery of *Saccharum* patches was evident by the stark contrast between the green colour of *Saccharum* and the surrounding black/burned landscape - further emphasised by the particularly hot and dry summer of 2022."

- Line 226: after 12th August, which other species appeared? In which order? How much cover after a period of time (if you have this data, I think you have it until 22nd May)?

We have these data, but we think that they are not relevant for the scopes of this paper.

- Line 229: How do you know that the regrowth is mostly Saccharum biflorum? With the drone images? Explain briefly.

We will explain briefly

Discussion

-Line 189: how did you decide the threshold value of 0.19? Do you have references from other similar studies to know which threshold do they use?

We agree this should have been better clarified. We actually started from the widely used definition of burns severity classes from Keeley (2009): https://pubs.usgs.gov/publication/70032718

We used then the following reference to obtain our first map with hard classes (below, not reported in the paper)

| | Severity Level | dNBR Range (scaled by $10^3$) | dNBR Range (not scaled) |
|---|---|---|---|
| | Enhanced Regrowth, high (post-fire) | -500 to -251 | -0.500 to -0.251 |
| | Enhanced Regrowth, low (post-fire) | -250 to -101 | -0.250 to -0.101 |
| | Unburned | -100 to +99 | -0.100 to +0.99 |
| | Low Severity | +100 to +269 | +0.100 to +0.269 |
| | Moderate-low Severity | +270 to +439 | +0.270 to +0.439 |
| | Miderate-high Severity | +440 to +659 | +0.440 to +0.659 |
| | High Severity | +660 to +1300 | +0.660 to +1.300 |

[Figure]

As it can be seen, false alarms appear in the urban area and in the west side of the island. As the division in classes of damage should be adapted to the case at hand, we considered that a conservative threshold should be applied, in order to identify an area which was damaged for certain. The value in the middle of the "Low Severity" class and rounded to the second decimal digit, 0.19, was selected as the lowest value without visible false alarms in the results.

- Line 243: You state that the fire affected strongly to the vegetation of the island, bu,t in a few days, the vegetation consisting in Saccharum biflorum recovered, then, the effect on the vegetation cover was only momentaneous and, at the short scale, the effect was not strong. I would highlight the strong recovery capacity of this species. In terms of the effect on the vegetation, I would speak about the loss in area of native vegetation, due to replacement of *S. biflorum* caused by fire.

Does this alien species appear in abandoned agricultural fields only, not being able to enter the areas with native vegetation?

We will provide additional details on this points, including the loss in area of native vegetation

- Line 247: Was all the population of Cytisus aeolicus destroyed. How much remained?

Fortunately, only a small part of the population of Cytisus aeolicus went burned, corresponding to 3% of its area of occupancy (as calculated by Zaia et al. 2020).

- Line 254: It would be good to state here or in methods the frequency of the fire in this area. Do the zones with native vegetation have a different fire frequency than areas with S. biflorum?

Unfortunately, we cannot add details on this point, due to the lack of details on the fire frequency. However, we do not believe that native vegetation has a different fire frequency than areas with Saccharum biflorum.

- Line 275: You would need to give more evidence or an hypothesis explaining what conditions could lead to this replacement.

This is done in lines 273-274: "There is no data on the longevity of Saccharum rhizomes and related senescence processes, nor on the effects of volcanic ash deposition on rhizome burial"

Conclusions

- Line 293: I would not put this in conclusions but in discussion

We agree.

- Line 302: Although S. biflorum avoids erosion, it favours fire. Would not be better to directly sowing the native species or putting seedlings? And prepare the soil for them.

Yes, of course. But in sloping sites Saccharum recovers (and grows) faster than the native vegetation that, in the first stages after fire, is dominated by annual plants and young seedlings of perennial plants, with limited soil retention capacity.

The study is not put in a broader context, comparing its results with similar studies in other Mediterranean areas or around the world. They could also compare the situation in Stromboli with cases in other islands of the Archipelago that have been affected by fire. Following your suggestion, we will add a comparison to other studies in the discussion.

**Citation**: https://doi.org/10.5194/bg-2023-19-RC3

---

## Author Response (AR2)

Dear Kirsten,

many thanks for your feedback on our manuscript.

All of your minor comments have been acknowledged and the Introduction was revised according to your suggestions. All site description, incl. land-use history and current situation have been moved to the subsection "Study area" of the Methods section. To better streamline the contents and avoid lengthy paragraphs, we added a subsection "Target species", focused on Saccharum biflorum.

The author team holds the journal Biogeoscience in high regard, both for the transparency of the review process and for being a society-owned, transdisciplinary journal with an excellent reputation. Thank you for taking the time to process our manuscript!

Point-by-point reply

1. Introduction needs to summarize the state-of-the-art on the research subject/topic. In this case, fire ecology and fire effects in the context of island ecology and introduced plant species in abandoned field. It finishes with summarizing the objectives of this study, optionally providing an overview of how the manuscript is structured. Please revise the introduction section accordingly and move all site description, incl. land-use history and current situation to the subsection Study area of the Methods section.

The Introduction was revised according to your suggestions. All site description, incl. land-use history and current situation have been moved to the subsection "Study area" of the Methods section. To better streamline the contents and avoid lengthy paragraphs, we added a subsection "Target species", focused on *Saccharum biflorum*.

Minor comments:

1. Line 85: "Despite the occurrence of some natural factors favouring fires, most of them are ignited by men through carelessness or", change here to "ignited by humans".
done

2. Address gender-neutral language (cross-check throughout the manuscript, e.g. line 413)
done

3. find a more understandable term for "paroxysmal activity" in line 119
done